# PRO-MOF: Policy Optimization with Universal Atomistic Models for Controllable MOF Generation

**Zicheng Liu**[1,2]   **Beng Fei**[3,4]   **Di Huang**[1,2,4*]

[1]BCC Lab, Hangzhou International Innovation Institute, Beihang University, Hangzhou, China
[2]School of Computer Science and Engineering, Beihang University, Beijing, China
[3]Chinese University of Hong Kong, Hong Kong, China
[4]Shenzhen Loop Area Institute, Shenzhen, China

## Abstract

Generating physically stable and novel metal-organic frameworks (MOFs) for inverse design that meet specific performance targets is a significant challenge. Existing generative models often struggle to explore the vast chemical and structural space effectively, leading to suboptimal solutions or mode collapse. To address this, we propose PRO-MOF, a hierarchical reinforcement learning (HRL) framework for controllable MOF generation. Our approach decouples the MOF design process into two policies: a high-level policy for proposing chemical building blocks and a low-level policy for assembling their 3D structures. By converting the deterministic Flow Matching model into a Stochastic Differential Equation (SDE), we enable the low-level policy to perform compelling exploration. The framework is optimized in a closed loop with high-fidelity physical reward signals provided by a pre-trained universal atomistic model (UMA). Furthermore, we introduce a Pass@K Group Relative Policy Optimization (GRPO) scheme that effectively balances exploration and exploitation by rewarding in-group diversity. Experiments on multiple inverse design tasks, such as maximizing CO2 working capacity and targeting specific pore diameters, show that PRO-MOF significantly outperforms existing baselines, including diffusion-based methods and genetic algorithms, in both success rate and the discovery of top-performing materials. Our work demonstrates that hierarchical reinforcement learning combined with a high-fidelity physical environment is a powerful paradigm for solving complex material discovery problems.

## 1 Introduction

Metal-Organic Frameworks (MOFs) represent a frontier in materials science, offering unprecedented structural and chemical tunability. Their vast internal surface areas and customizable pore environments make them exceptional candidates for critical applications such as carbon capture (Qian et al., 2020), gas storage, and catalysis. However, the combinatorial explosion of possible building blocks and topologies creates a design space of astronomical scale, rendering exhaustive exploration through traditional experimental or computational methods intractable. This challenge has catalyzed the development of generative models, which aim to navigate this vast space and accelerate the discovery of novel materials. Recent advances, particularly coarse-grained diffusion models like DiffCSP (Jiao et al., 2024), MOFDiff (Fu et al., 2023), and flexible assembly frameworks like MOFFlow-2 (Kim et al., 2025a), have marked significant progress. These models can generate MOFs with novel and geometrically plausible structures, moving beyond the limitations of predefined templates. They excel at learning the complex statistical distributions of atomic arrangements and molecular connectivity from large datasets. However, their success in capturing geometric realism often masks a critical deficiency: a lack of inherent physical viability.

---

*Corresponding author: `dhuang@buaa.edu.cn`

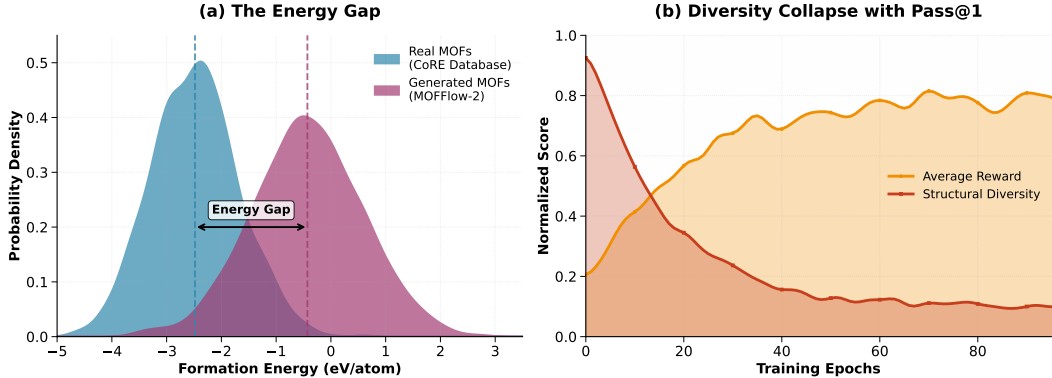

Figure 1: The core motivations for PRO-MOF. (a) The Energy Gap: The distribution of formation energies for MOFs from a baseline generator (e.g., MOFFlow-2) is significantly higher and broader than that of real materials, indicating widespread physical instability. (b) Diversity Collapse: Standard reinforcement learning with a Pass@1 reward function successfully increases the average reward but at the cost of a drastic reduction in structural diversity, leading to mode collapse.

This "physical reality gap" is a fundamental barrier to practical materials discovery. As we demonstrate, a significant fraction of structures generated by state-of-the-art models, while appearing structurally sound, are physically unstable. We evaluated the formation energies of thousands of generated structures using a universal machine learning interatomic potential, UMA (Wood et al., 2025), which serves as a high-fidelity surrogate for DFT calculations (Garrity et al., 2014). The results, shown in Figure 1(a), reveal a stark disparity: the energy distribution of generated MOFs is dramatically shifted towards higher, less stable states compared to the distribution of real-world, synthesizable materials.

A natural solution to this problem is to integrate physical realism directly into the training loop using reinforcement learning (RL), with a fast and accurate model like UMA providing the reward signal. However, a naive implementation of RL creates a new, more subtle trap. Standard policy optimization, which rewards the generation of a single successful sample (a *Pass@1* reward), inherently favors exploitation over exploration (Shao et al., 2024). The generator quickly learns to exploit a few "safe" regions of the chemical space where it can reliably produce stable structures, ceasing to explore novel topologies. This leads to a rapid collapse in the diversity of generated candidates, a phenomenon known as mode collapse (Figure 1(b)). For a task whose ultimate goal is the discovery of *new* materials, this convergence to a local optimum is a critical failure.

In this work, we introduce **PRO-MOF** (Policy and Reward Optimized MOFs), a holistic framework that addresses both the physical reality gap and the diversity collapse problem. PRO-MOF synergistically combines a state-of-the-art flow-based MOF generator with a universal atomistic model (UMA) inside a novel policy optimization loop. To overcome the exploration-exploitation dilemma, we move beyond simplistic rewards and implement a *Pass@K* policy optimization strategy, inspired by recent advances in training large reasoning models (Chen et al., 2025). By rewarding the generator for producing at least one successful candidate within a batch of $k$ diverse attempts, our framework intrinsically incentivizes exploration. This encourages the model to discover multiple, distinct islands of stability across the vast MOF landscape. Our main contributions are:

- We propose PRO-MOF, the first framework to successfully employ online reinforcement learning for controllable, de novo MOF generation by leveraging a universal atomistic model as a high-fidelity environment.

- We introduce a Pass@K-inspired reward and advantage estimation scheme to the materials domain, which explicitly promotes structural diversity and effectively mitigates the mode collapse problem inherent in standard RL approaches.

- We demonstrate through extensive experiments that PRO-MOF significantly outperforms existing methods in generating diverse, stable, and high-performing MOFs for targeted inverse design tasks, paving the way for more efficient computational materials discovery.

## 2 PRELIMINARIES

**Notation for MOF Representation**  We adopt the flexible, coarse-grained representation of MOFs introduced by Kim et al. (2025b;a), which models a structure based on its constituent building blocks. An entire MOF structure with $N$ atoms is denoted by the tuple $\mathcal{S} = (\mathcal{B}_{3D}, \boldsymbol{q}, \boldsymbol{\tau}, \boldsymbol{\phi}, \boldsymbol{\ell})$. This tuple consists of: a set of $M$ 3D building blocks, $\mathcal{B}_{3D} = \{\mathcal{C}^{(m)}\}_{m=1}^M$, where each block $\mathcal{C}^{(m)}$ is defined by its atom types and local coordinates; the rigid-body rotations $\boldsymbol{q} = \{q^{(m)} \in SO(3)\}_{m=1}^M$ for each building block; the rigid-body translations $\boldsymbol{\tau} = \{\tau^{(m)} \in \mathbb{R}^3\}_{m=1}^M$ for each block; the internal torsion angles $\boldsymbol{\phi} = \{\phi^{(m)} \in SO(2)^{P_m}\}_{m=1}^M$ for the $P_m$ rotatable bonds within each block to capture linker flexibility; and the global lattice parameters $\boldsymbol{\ell} = (a, b, c, \alpha, \beta, \gamma)$ defining the periodic unit cell. The complete atomic coordinates $\boldsymbol{X} \in \mathbb{R}^{N \times 3}$ can be reconstructed by applying these transformations to the local coordinates of the building blocks.

**Generative Modeling with Flow Matching**  Flow Matching (FM) (Liu et al., 2023) is a powerful simulation-free framework for training Continuous Normalizing Flows (CNFs) (Lipman et al., 2023). The core idea is to learn a time-dependent vector field (or velocity field) $\boldsymbol{v}_\theta(\boldsymbol{x}_t, t)$ that transports samples from a simple prior distribution $p_1$ (e.g., Gaussian noise) to a complex target data distribution $p_0$. This is achieved by regressing the model's predicted velocity against a target velocity field defined along a path between a noise sample $\boldsymbol{x}_1$ and a real data sample $\boldsymbol{x}_0$. For generation, one samples $\boldsymbol{x}_1 \sim p_1$ and solves the ordinary differential equation (ODE) $\frac{d\boldsymbol{x}_t}{dt} = \boldsymbol{v}_\theta(\boldsymbol{x}_t, t)$ from $t = 1$ to $t = 0$. This process is deterministic and typically highly efficient, allowing for high-quality sample generation in few steps. However, its deterministic nature precludes the stochastic exploration required by online RL algorithms.

**Policy Optimization with GRPO**  Group Relative Policy Optimization (GRPO) (Shao et al., 2024) is a memory-efficient, value-function-free online RL algorithm that has proven effective for optimizing large generative models (Liu et al., 2025). The generative process is framed as a Markov Decision Process (MDP). At each optimization step, the policy $\pi_\theta$ generates a group of $G$ samples. A reward function provides a scalar reward $R_i$ for each completed sample. Instead of learning a value function, GRPO computes the advantage for each sample *relative to its peers within the group*:

$$\hat{A}_i = \frac{R_i - \mu_R}{\sigma_R + \epsilon}, \tag{1}$$

where $\mu_R$ and $\sigma_R$ are the mean and standard deviation of rewards within the group. This relative advantage is then used in a clipped policy gradient objective to update the policy parameters $\theta$. A key requirement for GRPO is a stochastic policy that can generate diverse samples within a group to ensure a meaningful reward distribution and effective exploration.

## 3 PRO-MOF: A FRAMEWORK FOR CONTROLLABLE MOF GENERATION

Our methodology, PRO-MOF, is designed to overcome two fundamental challenges in de novo materials generation: the *physical reality gap* and the *exploration-exploitation dilemma*. PRO-MOF addresses these by formulating the generation process as a **hierarchical policy optimization problem**. We co-design the MOF's chemical composition and its geometric structure by training two specialized policies in a closed loop, guided by rewards from a universal atomistic model. This approach ensures that both the selection of chemical building blocks and their 3D assembly are simultaneously optimized toward generating physically viable and high-performing structures.

### 3.1 A HIERARCHICAL FRAMEWORK FOR DE NOVO MOF GENERATION

We conceptualize the de novo MOF generation process as a two-stage hierarchy, building upon the paradigm of MOFFlow-2 (Kim et al., 2025a). This structure decouples the vast, discrete chemical space from the high-dimensional, continuous geometric space, allowing each policy to specialize.

**High-Level Policy $\pi_\theta^{\text{chem}}$ (The Chemist)**  The first stage is governed by a high-level policy, an autoregressive Transformer model. This policy operates in the *chemical space*. Its action, $a_{\text{chem}}$, is to generate a canonicalized sequence of building blocks represented as SMILES strings. Following

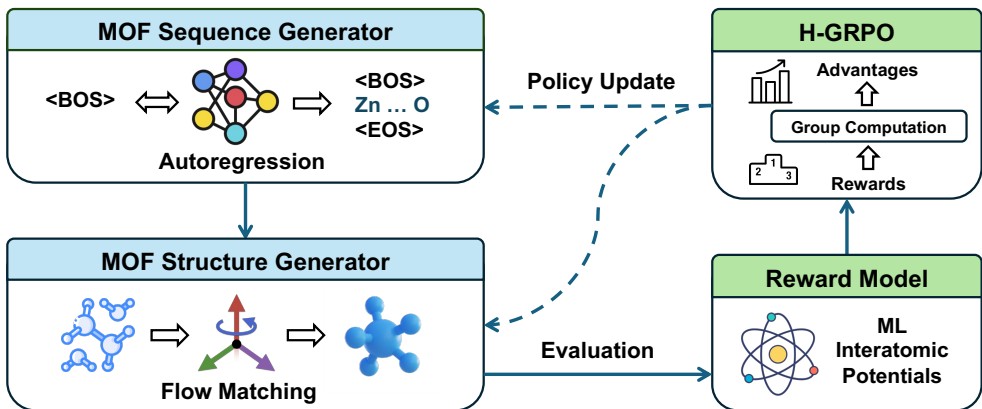

Figure 2: The PRO-MOF Hierarchical Reinforcement Learning Framework. The generation process is decoupled into two policies: a high-level MOF Sequence Generator that proposes chemical building blocks, and a low-level MOF Structure Generator that assembles them. A universal Reward Model (a Machine Learning Interatomic Potential, UMA) evaluates the final structure's quality. The reward signal is then used by the H-GRPO module to compute advantages and perform a policy update, closing the loop to optimize both chemical and structural generation simultaneously.

Kim et al. (2025a), the sequence is structured as: $\mathcal{B}_{2D} = $ <BOS> $m_1.m_2...$ <SEP> $o_1.o_2...$ <EOS>, where metal clusters ($m_i$) precede organic linkers ($o_j$), and components within each group are sorted by molecular weight. This canonical representation defines the set of 2D chemical building blocks, $\mathcal{B}_{2D}$. The policy's objective is to propose building blocks with a high potential for forming stable, high-performing MOFs.

**Low-Level Policy $\pi_\phi^{\text{geom}}$ (The Structural Engineer)**   The second stage is controlled by a low-level policy, a conditional flow matching model. This policy operates in the *geometric space*. It receives the building blocks $a_{\text{chem}}$ and first initializes their 3D structures, $\mathcal{B}_{3D}$, using a pre-defined library for metal clusters and cheminformatics tools, RDKit (Landrum, 2006–), for organic linkers, as done in MOFFlow-2 (Kim et al., 2025a). Given $\mathcal{B}_{3D}$, its action, $a_{\text{geom}}$, is to determine the optimal 3D assembly. This action corresponds to generating the continuous parameters for rigid-body translations ($\tau$) and rotations ($q$), flexible linker torsions ($\phi$), and the global lattice ($\ell$). The policy is a non-equivariant Transformer architecture that learns a time-dependent vector field $v_\phi$ to transport noise to the correct structural parameters.

The final MOF structure $\mathcal{S}$ is a deterministic function of both actions, $\mathcal{S} = f(a_{\text{chem}}, a_{\text{geom}})$. A naive approach of training these models separately via maximum likelihood leads to the problems outlined in Section 1. Our solution is to train them jointly using a hierarchical reinforcement learning scheme.

## 3.2 HIERARCHICAL POLICY OPTIMIZATION WITH H-GRPO

We frame the co-design task within a hierarchical reinforcement learning (HRL) framework. The high-level policy proposes a "sub-task" (a set of building blocks), and the low-level policy attempts to solve it (find a stable assembly). The final reward, derived from the all-atom structure, is then used to update both policies, addressing the critical credit assignment problem. In RL training, we adopt the pre-trained models from Kim et al. (2025a).

**Policy Optimization Loop**   The core of PRO-MOF is a closed loop (Figure 2) where the two policies are optimized iteratively. For a given target property $c$, the high-level policy $\pi_\theta^{\text{chem}}$ first generates a MOF sequence $a_{\text{chem}}$. This action is passed to the low-level policy $\pi_\phi^{\text{geom}}$, which then explores the geometric space by stochastically generating a diverse group of $k$ possible 3D structures $\{\mathcal{S}_1, \ldots, \mathcal{S}_k\}$. A universal atomistic model, UMA, evaluates all $k$ structures, providing the rewards.

**Hierarchical Credit Assignment**    The key to our HRL framework is how rewards are distributed to solve the credit assignment problem. The rewards $\{R_1, \ldots, R_k\}$ for the individual 3D structures are used to directly update the low-level (geometric) policy $\pi_\phi^{\text{geom}}$, using the Pass@K GRPO scheme described in Section 3.3. Crucially, the reward for the high-level (chemical) action $a_{\text{chem}}$ is defined as the *best possible outcome* achievable with the chosen building blocks. We thus define the high-level reward as $R_{\text{chem}} = \max(R_1, \ldots, R_k)$. This signal informs the high-level policy about the "potential" of the building blocks it generated. This scalar reward is then used to update $\pi_\theta^{\text{chem}}$'s parameters using a clipped policy gradient objective, similar to the one used for the low-level policy but adapted for a discrete action space.

## 3.3    Geometric Policy Optimization via Pass@K GRPO

Optimizing the low-level geometric policy $\pi_\phi^{\text{geom}}$ requires several technical innovations to enable stable and effective online reinforcement learning.

### 3.3.1    Stochastic Exploration via SDE Sampling

To enable exploration, we convert the deterministic ODE of the flow matching model into an equivalent Stochastic Differential Equation (SDE), following Flow-GRPO (Liu et al., 2025). The probability flow ODE, $d\boldsymbol{x}_t = \boldsymbol{v}_t(\boldsymbol{x}_t)dt$, is transformed into a reverse-time SDE. For a rectified flow where $p_t$ is an interpolation of $p_0$ and $p_1$, the score term $\nabla \log p_t(\boldsymbol{x}_t)$ can be related to the velocity field $\boldsymbol{v}_t$. This results in the following tractable SDE:

$$d\boldsymbol{x}_t = \left[ \boldsymbol{v}_\phi(\boldsymbol{x}_t, t) + \frac{\sigma_t^2}{2t}(\boldsymbol{x}_t + (1-t)\boldsymbol{v}_\phi(\boldsymbol{x}_t, t)) \right] dt + \sigma_t d\boldsymbol{w}, \tag{2}$$

where $\sigma_t$ is a time-dependent diffusion coefficient that controls the level of stochasticity and $d\boldsymbol{w}$ is a standard Wiener process. This transformation allows the geometric policy to sample stochastically from a Gaussian policy $\pi_\phi(x_{t-1}|x_t, c)$, a prerequisite for RL-based exploration. The full derivation is provided in Appendix A.

### 3.3.2    Multi-Objective Reward Function with UMA

We leverage UMA (Wood et al., 2025), a state-of-the-art universal machine learning interatomic potential (MLIP), to define a multi-objective reward function. For a generated structure $\mathcal{S}$, the total reward $R_{\text{total}}$ is a weighted sum:

$$R_{\text{total}}(\mathcal{S}, c) = w_{\text{stability}} R_{\text{stability}}(\mathcal{S}) + w_{\text{property}} R_{\text{property}}(\mathcal{S}, c). \tag{3}$$

The Stability Reward ($R_{\text{stability}}$) is derived from the UMA-evaluated potential energy of the relaxed structure, $E_{\text{UMA}}(\mathcal{S})$, defined as $R_{\text{stability}} = -\log(E_{\text{UMA}}(\mathcal{S}) - E_{\text{min}})$, encouraging physically viable, low-energy configurations. The Property Matching Reward ($R_{\text{property}}$) measures how well the structure's properties, also predicted by UMA, match the target condition $c$. Further details on UMA are in Appendix B.

### 3.3.3    Pass@K Advantage Estimation

A naive application of RL can lead to mode collapse, where the policy repeatedly generates a single high-reward structure. To combat this, we adopt a Pass@K strategy (Chen et al., 2025) within the GRPO framework. The geometric policy generates a batch of $k$ candidates $\{\mathcal{S}_1, \ldots, \mathcal{S}_k\}$ for a given $a_{\text{chem}}$. The rewards $\{R_1, \ldots, R_k\}$ are used to compute the group-relative advantage for each sample $l \in \{1, \ldots, k\}$:

$$\hat{A}_l = \frac{R_l - \mu_R}{\sigma_R + \epsilon}. \tag{4}$$

The policy $\pi_\phi$ is then updated using a clipped objective based on these advantages. This group-based advantage intrinsically incentivizes the policy to explore diverse yet successful geometric configurations, as a high reward for one sample does not suppress the learning signal for other promising candidates.

---

**Algorithm 1** PRO-MOF Hierarchical Training Loop

---

1: Initialize high-level policy $\pi_\theta^{\text{chem}}$ and low-level policy $\pi_\phi^{\text{geom}}$ from pre-trained MLE models.
2: Initialize reference policies $\pi_{\theta,\text{ref}} \leftarrow \pi_\theta^{\text{chem}}$, $\pi_{\phi,\text{ref}} \leftarrow \pi_\phi^{\text{geom}}$.
3: Set total warmup iterations $I_{\text{warmup}}$.
4: **for** training iteration $i = 1, 2, \ldots$ **do**
5:      Sample a batch of target properties $\{c_1, \ldots, c_B\}$.
6:      Compute annealing weight $w_{\text{anneal}}(i) = \min(1, i/I_{\text{warmup}})$.
7:      *// === High-Level Policy Rollout and Low-Level Execution ===*
8:      **for** each target property $c_j$ **do**
9:          Generate a MOF sequence $a_{\text{chem},j} \sim \pi_\theta^{\text{chem}}(\cdot|c_j)$.
10:         Generate $k$ structures $\{\mathcal{S}_{j,1}, \ldots, \mathcal{S}_{j,k}\} \sim \pi_\phi^{\text{geom}}(\cdot|a_{\text{chem},j}, c_j)$ using SDE sampler (Eq. 2).
11:         Evaluate rewards $\{R_{j,1}, \ldots, R_{j,k}\}$ for all $k$ structures using UMA and apply annealing (Eq. 5).
12:         Compute high-level reward $R_{\text{chem},j} = \max(R_{j,1}, \ldots, R_{j,k})$.
13:      **end for**
14:      *// === Policy Updates with KL Regularization ===*
15:      Update $\pi_\theta^{\text{chem}}$ using rewards $\{R_{\text{chem},j}\}$ and advantage $\hat{A}_j$ via a clipped objective:
16:      $\mathcal{L}_\theta = \mathbb{E}[\min(r_j(\theta)\hat{A}_j, \text{clip}(r_j(\theta), 1-\epsilon, 1+\epsilon)\hat{A}_j) - \beta D_{KL}(\pi_\theta \| \pi_{\theta,\text{ref}})]$
17:      Update $\pi_\phi^{\text{geom}}$ using all rewards $\{\{R_{j,l}\}\}$ and advantages $\{\{\hat{A}_{j,l}\}\}$ via GRPO objective:
18:      $\mathcal{L}_\phi = \mathbb{E}[\min(r_{j,l}(\phi)\hat{A}_{j,l}, \text{clip}(r_{j,l}(\phi), 1-\epsilon, 1+\epsilon)\hat{A}_{j,l}) - \beta D_{KL}(\pi_\phi \| \pi_{\phi,\text{ref}})]$
19: **end for**

---

### 3.3.4 REWARD ANNEALING FOR STABLE OPTIMIZATION

A critical challenge arises as the policy $\pi_\phi^{\text{geom}}$ in its early training stages may produce non-physical inter-block configurations that are out-of-distribution for UMA, leading to noisy reward signals and unstable gradients. To address this, we introduce a **reward annealing** strategy. We modulate the total reward with a weight $w_{\text{anneal}}(i)$ that increases with the training iteration $i$:

$$R_{\text{effective}}(i) = w_{\text{anneal}}(i) \cdot R_{\text{total}}(\mathcal{S}, c), \quad \text{where} \quad w_{\text{anneal}}(i) = \min\left(1, \frac{i}{I_{\text{warmup}}}\right). \tag{5}$$

This curriculum, where $I_{\text{warmup}}$ is a hyperparameter, ensures that the policy initially receives a gentle learning signal from the pre-trained MLE objective, preventing destabilization from noisy rewards and promoting stable convergence as it begins to explore.

## 4 RELATED WORK

**Generative Models for MOF Design.** Generative modeling for Metal-Organic Frameworks (MOFs) presents unique challenges compared to smaller inorganic crystals, primarily due to the large size of MOF unit cells, which often contain hundreds of atoms (Cao et al., 2022; Badrinarayanan et al., 2025). To tackle this complexity, recent methods have employed hierarchical or coarse-grained approaches. For example, MOFDiff (Fu et al., 2023) first generates a coarse-grained representation of the structure via a diffusion model and then decodes it into an all-atom structure using a predefined library of building blocks extracted from the training data. Similarly, MOF-Flow (Kim et al., 2025b) simplifies the problem by treating building blocks as rigid bodies, learning their relative rotations and translations rather than predicting individual atomic coordinates, but this requires access to the 3D structures of the building blocks beforehand.

**Universal Atomistic Models as Simulation Engines.** A prevailing trend in molecular modeling is the development of universal, data-driven models that can act as general-purpose simulation engines. A key architectural consideration in these models is the enforcement of SE(3) or E(3) equivariance to respect the physical symmetries of 3D space. While equivariant generative models have shown promising results on many molecular tasks (Corso et al., 2023; Jing et al., 2022; Jumper et al., 2021; Zeni et al., 2025), a growing body of work questions whether strict equivariance is essential for achieving state-of-the-art performance (Wang et al., 2024). Notably, leading

models for biomolecular complexes and crystal structures, such as AlphaFold3 (Abramson et al., 2024), Boltz-1 (Wohlwend et al., 2024), Proteína (Geffner et al., 2025), Orb (Neumann et al., 2024), and ADiT (Joshi et al., 2025), have achieved exceptional results using non-equivariant architectures, like the Transformer. Following this successful trend, PRO-MOF adopts a non-equivariant Transformer architecture, demonstrating that it can effectively learn the complex potential energy surface of MOFs without built-in geometric constraints.

**Reinforcement Learning for Generative Model Refinement.** While generative models are powerful for learning the underlying distribution of a dataset, they do not inherently optimize for specific functional properties. Reinforcement Learning (RL) has emerged as a potent strategy to fine-tune generative models for goal-oriented de novo design (Popova et al., 2018). In this paradigm, a pretrained generative model acts as a policy that produces candidate molecules or materials. A reward function, often defined by a fast property predictor or a surrogate model, then provides feedback to guide the generator towards regions of the chemical space with more desirable properties. This approach has been widely and successfully applied in drug discovery to optimize for properties like binding affinity and synthesizability (Kim et al., 2021; Schwaller et al., 2019; Olivecrona et al., 2017). In materials science, RL offers a promising pathway to steer the generation of novel structures, such as MOFs, towards enhanced performance for specific applications, moving beyond simple reproduction of known structural motifs (Kim et al., 2020).

## 5 EXPERIMENTS

We design a suite of experiments to rigorously evaluate PRO-MOF and demonstrate its superiority over existing methods. Our evaluation focuses on the model's ability to perform *controllable inverse design*, a task that requires generating MOFs that are not only physically stable and novel but also satisfy specific, user-defined property targets.

### 5.1 EXPERIMENTAL SETUP

**Datasets and Base Models.** We use the hypothetical MOF database from Kim et al. (2025a) for tuning the pre-trained generative model; the details of dataset processing are described in Appendix D. The base generator is a conditional version of the MOFFlow-2 architecture, pre-trained to generate MOFs conditioned on a vector of their chemical properties. The reward function and environment for RL fine-tuning are powered by the pre-trained UMA-S model (Wood et al., 2025), which provides rapid and accurate energy evaluations.

**Inverse Design Tasks.** We evaluate all models on three scientifically relevant inverse design tasks. The first is maximizing the $CO_2$ working capacity, a benchmark task for carbon capture applications directly comparable to Fu et al. (2023). The evaluation pipeline is also adopted from Fu et al. (2023), which means we are tuning the model using simulated data. The second task involves targeting specific pore diameters, crucial for gas separation, where the objective is to generate stable MOFs with a pore limiting diameter (PLD) within a narrow range (e.g., $6.0 \pm 0.2$Å). The final task is a pure exploration challenge: discovering ultra-stable novel topologies by finding MOFs with the lowest possible formation energies, without other property constraints.

| Method | Max CO2 Cap. | | Target PLD | | Min Energy | |
|---|---|---|---|---|---|---|
| | Success Rate | Top-1 | Success Rate | Top-1 | Success Rate | Top-1 |
| MOFDiff (Latent Opt.) | $2.1_{\pm0.3}\%$ | $4.9_{\pm0.2}$ | $0.8_{\pm0.2}\%$ | $6.5_{\pm0.2}$ Å | $1.5_{\pm0.3}\%$ | $-0.95_{\pm0.04}$ eV |
| MOFFlow-2 (S&F) | $3.5_{\pm0.4}\%$ | $5.1_{\pm0.2}$ | $1.2_{\pm0.3}\%$ | $5.9_{\pm0.2}$ Å | $2.8_{\pm0.4}\%$ | $-1.02_{\pm0.04}$ eV |
| MOFFlow-2 (Release) | $4.0_{\pm0.4}\%$ | $5.2_{\pm0.2}$ | $2.2_{\pm0.3}\%$ | $5.9_{\pm0.2}$ Å | $3.6_{\pm0.4}\%$ | $-1.05_{\pm0.04}$ eV |
| GA+UMA | $6.2_{\pm0.5}\%$ | $5.4_{\pm0.2}$ | $2.5_{\pm0.4}\%$ | $6.1_{\pm0.2}$ Å | $5.5_{\pm0.5}\%$ | $-1.15_{\pm0.03}$ eV |
| PRO-MOF (Pass@1) | $8.1_{\pm0.6}\%$ | $5.6_{\pm0.2}$ | $3.1_{\pm0.4}\%$ | $6.0_{\pm0.1}$ Å | $7.2_{\pm0.6}\%$ | $-1.21_{\pm0.03}$ eV |
| **PRO-MOF (Pass@3)** | $\mathbf{10.3_{\pm0.7}\%}$ | $\mathbf{5.9_{\pm0.2}}$ | $\mathbf{7.8_{\pm0.5}\%}$ | $\mathbf{6.0_{\pm0.1}}$ **Å** | $\mathbf{12.4_{\pm0.8}\%}$ | $\mathbf{-1.35_{\pm0.02}}$ **eV** |

Table 1: Main results on inverse design tasks. We report the Success Rate (%) and the Top-1 property value achieved within a fixed computational budget. PRO-MOF consistently discovers more and better candidates across all tasks.

**Baselines.** We compare PRO-MOF against a comprehensive set of strong baselines. This includes the latent space optimization approach from MOFDiff (Fu et al., 2023), representing the state-of-the-art in diffusion-based design. We also establish a non-RL baseline, MOFFlow-2 (Sample & Filter), which involves generating a large number of samples from the pre-trained generator and using UMA for post-hoc screening. To isolate the benefits of our exploration strategy, we include an ablation of our own method, PRO-MOF (Pass@1), which uses a standard Pass@1 reward scheme. Finally, we compare against a classic and powerful optimization method, a Genetic Algorithm that evolves a population of MOFs using UMA as the fitness function (GA+UMA).

**Evaluation Metrics.** We grant all methods an identical computational budget, defined as 10,000 UMA evaluations. Performance is measured by a set of key metrics. The *Success Rate* is the number of generated MOFs that are both physically stable (low energy after relaxation) and satisfy the task-specific property target. *Top-K Performance* reports the average property value of the top-K successful candidates found. To quantify exploration, we measure the chemical and structural *Diversity* of the successful candidates using the average Tanimoto distance of their structural fingerprints. Finally, we report standard *Validity* and *Novelty* metrics (Kim et al., 2025a) to ensure that the generated structures are both chemically sound and novel.

## 5.2 MAIN RESULTS

As shown in Table 1, PRO-MOF with the Pass@K strategy consistently and significantly outperforms all baseline methods across every task. Notably, it not only achieves the highest success rate—discovering valid, on-target candidates more efficiently—but also finds superior materials, as indicated by the Top-1 performance values. The comparison against the Pass@1 version of our own framework highlights the critical role of the Pass@K strategy in navigating the complex design space to find better solutions.

## 5.3 ABLATION STUDIES

To dissect the contributions of our framework's key components, we perform a detailed ablation study on Task 2 (Targeting Pore Diameters). Table 2 demonstrates that removing either the Pass@K reward scheme (reverting to Pass@1) or the UMA reward model (using a simpler proxy) results in a substantial drop in performance, particularly in success rate and diversity. This confirms that both the exploration-promoting RL strategy and the high-fidelity physical environment are essential for the success of PRO-MOF.

## 5.4 ANALYSIS OF GENERATION QUALITY AND DIVERSITY

We further analyze the behavior of our method during training. To directly assess the impact of our optimization, Figure 4 compares the formation energy distributions of generated MOFs.

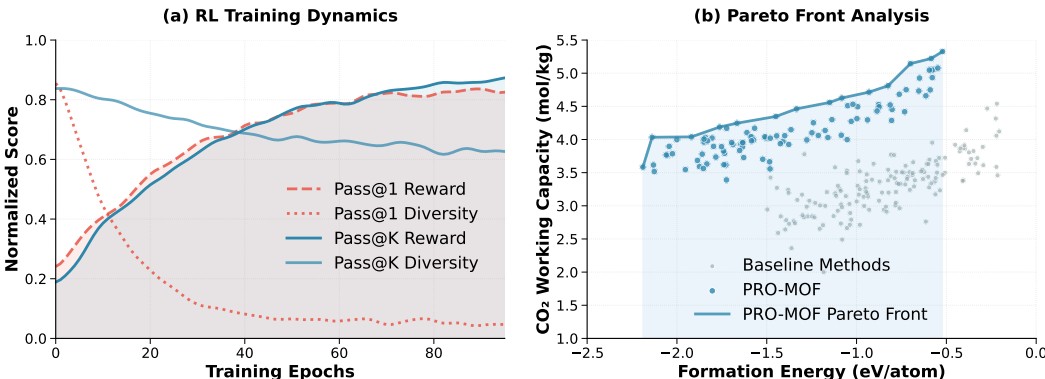

Figure 3: (a) Training curves comparing Pass@K and Pass@1 strategies. Pass@K maintains high diversity while achieving superior rewards. (b) Left: Pareto plot of stability vs. CO2 capacity for generated MOFs, showing PRO-MOF discovers a better frontier. Right: Visualization of novel, high-performing MOF structures discovered by PRO-MOF.

| Method | Success Rate (%) ↑ | Top-1 Value (Best is 6.0Å) | Diversity ↑ |
|---|---|---|---|
| **PRO-MOF (Full Method)** | **7.8** | **6.0 Å** | **0.65** |
| w/o Pass@K (uses Pass@1) | 3.1 | 6.0 Å | 0.31 |
| w/o UMA | 1.9 | 6.4 Å | 0.45 |

Table 2: Ablation study on the Targeted Pore Diameter task.

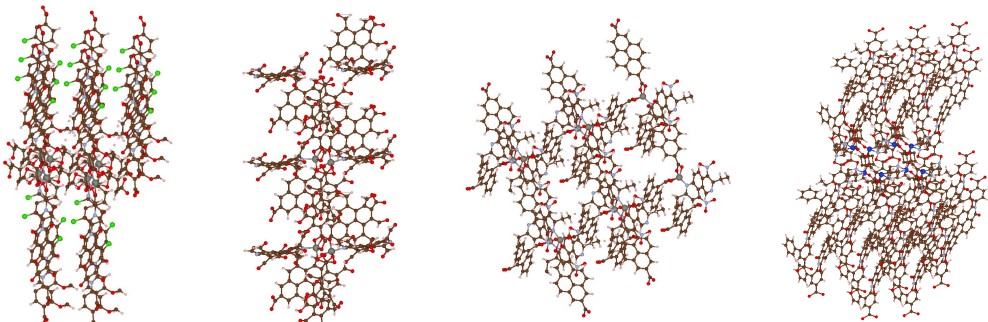

Figure 5: Visualizations of samples generated by PRO-MOF. Atom color code: Cu (blue), Zn (gray), O (red), N (purple), C (brown), H (white), Cl (green).

The original, pre-trained MOFFlow-2 model (purple distribution) yields structures with higher energies, showing a clear gap compared to the target distribution of real MOFs (blue). After optimization with PRO-MOF, the resulting energy distribution (green) shifts significantly towards lower energies, aligning much more closely with the real MOF data. This demonstrates that our method effectively guides the generator to explore more stable regions of the chemical space, discovering novel structures that are physically more viable.

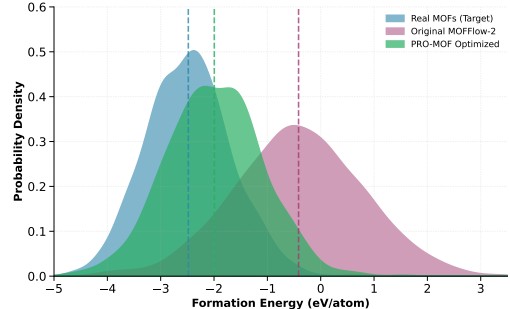

Figure 4: A comparison of the formation energy distributions for generated MOFs.

Figure 3(a) illustrates the trade-off between reward maximization and diversity preservation. The Pass@1 strategy increases the average reward but at the cost of a sharp decline in sample diversity, confirming the mode collapse behavior hypothesized in our introduction. In contrast, our Pass@K strategy maintains a high level of diversity training while still achieving a superior reward, demonstrating its effectiveness at balancing exploration and exploitation.

Figure 3(b) provides a qualitative view of the generated structures. The Pareto plot shows that PRO-MOF discovers a frontier of MOFs that are both more stable (lower energy) and higher performing (higher CO2 capacity) than those found by baselines. The visualizations showcase top-performing novel MOFs discovered by PRO-MOF, with unique topologies not present in the training data, confirming that our model is not merely memorizing but is capable of genuine innovation. Figure 5 provides further examples of the chemically diverse MOFs generated by our method, highlighting its ability to produce novel materials with intricate pore environments and connectivity.

## 6 CONCLUSION

In this work, we introduced PRO-MOF, a novel framework for the controllable inverse design of Metal-Organic Frameworks. Our approach tackles two of the most pressing challenges in generative materials science: the physical instability of generated structures and the diversity collapse induced by standard optimization techniques. By synergistically integrating a flow-based generative model, a universal atomistic model as a high-fidelity environment, and a Pass@K-driven reinforcement learning strategy, PRO-MOF creates a robust, closed-loop system for materials discovery.

**Limitations**  While PRO-MOF establishes a powerful new methodology, we acknowledge several limitations that open avenues for future research. First, the efficacy of our framework is fundamentally tied to the accuracy and domain coverage of the universal atomistic model. Any inaccuracies or "blind spots" in the UMA's potential energy surface could be exploited by the RL agent, leading to the generation of artifactually stable structures. Besides, our framework currently optimizes the assembly of pre-generated building blocks. A truly end-to-end approach would involve integrating chemical space exploration (i.e., generating SMILES strings for building blocks) into the RL loop.

## ACKNOWLEDGMENT

This work is partly supported by the Sino-French Research Partner Exchange Program under the PHC-CNC project (Artemis: An AI-Enhanced Approach to Waste Sorting and Decarbonization) and the Fundamental Research Funds for the Central Universities.

## ETHICS STATEMENT

Our research aims to accelerate the discovery of materials for beneficial societal applications, such as carbon capture. The work utilizes public datasets and open-source models, adhering to responsible scientific practices. While we acknowledge the potential for dual-use applications in any generative model, our methodology is focused on solving established problems in materials science. We intend to release our code to promote transparency and further beneficial research.

## REPRODUCIBILITY STATEMENT

Our methodology relies on public datasets and models, with all preprocessing steps (D) and hyperparameters (C) detailed in this work. We will also release the final trained model weights to allow for the full replication of our experimental results.

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

## A APPENDIX

## A DERIVATION OF SDE FOR STOCHASTIC SAMPLING

To enable stochastic exploration for reinforcement learning, we convert the deterministic Ordinary Differential Equation (ODE) used in flow matching into a corresponding Stochastic Differential Equation (SDE) that preserves the marginal probability densities $p_t(x)$ at all times $t \in [0, 1]$. This derivation follows the principles outlined in Liu et al. (2025) and Song et al. (2021).

The original generative process is defined by a probability flow ODE:

$$d\boldsymbol{x}_t = \boldsymbol{v}_t(\boldsymbol{x}_t)dt, \tag{6}$$

where $\boldsymbol{v}_t$ is the learned velocity field. The evolution of the probability density $p_t(\boldsymbol{x})$ under this ODE is given by the continuity equation:

$$\frac{\partial p_t(\boldsymbol{x})}{\partial t} = -\nabla \cdot (\boldsymbol{v}_t(\boldsymbol{x})p_t(\boldsymbol{x})). \tag{7}$$

We seek a corresponding reverse-time SDE of the form:

$$d\boldsymbol{x}_t = \boldsymbol{f}(\boldsymbol{x}_t, t)dt + g(t)d\boldsymbol{w}, \tag{8}$$

where $d\boldsymbol{w}$ is a standard Wiener process. The time evolution of its density is described by the Fokker-Planck equation:

$$\frac{\partial p_t(\boldsymbol{x})}{\partial t} = -\nabla \cdot (\boldsymbol{f}(\boldsymbol{x}_t, t)p_t(\boldsymbol{x})) + \frac{1}{2}g(t)^2\nabla^2 p_t(\boldsymbol{x}). \tag{9}$$

For the marginals to match, we must equate the right-hand sides of the continuity and Fokker-Planck equations. This gives the drift term $\boldsymbol{f}(\boldsymbol{x}_t, t) = \boldsymbol{v}_t(\boldsymbol{x}_t) - \frac{1}{2}g(t)^2\nabla \log p_t(\boldsymbol{x}_t)$.

For rectified flow, the path is a linear interpolation $x_t = (1 - t)x_0 + tx_1$. The score function $\nabla \log p_t(x)$ can be directly related to the velocity field $v_t(x)$ (Liu et al., 2025):

$$\nabla \log p_t(\boldsymbol{x}) = -\frac{\boldsymbol{x}}{t} - \frac{1-t}{t}\boldsymbol{v}_t(\boldsymbol{x}). \tag{10}$$

Substituting this into the drift term yields the final reverse-time SDE:

$$d\boldsymbol{x}_t = \left(\boldsymbol{v}_t(\boldsymbol{x}_t) - \frac{\sigma_t^2}{2}\left(-\frac{\boldsymbol{x}_t}{t} - \frac{1-t}{t}\boldsymbol{v}_t(\boldsymbol{x}_t)\right)\right)dt + \sigma_t d\boldsymbol{w}, \tag{11}$$

which simplifies to:

$$d\boldsymbol{x}_t = \left[\boldsymbol{v}_t(\boldsymbol{x}_t) + \frac{\sigma_t^2}{2t}(\boldsymbol{x}_t + (1-t)\boldsymbol{v}_t(\boldsymbol{x}_t))\right]dt + \sigma_t d\boldsymbol{w}. \tag{12}$$

This equation can be discretized using an Euler-Maruyama solver for practical implementation, providing the stochastic sampling mechanism required for GRPO.

## B UMA AS A UNIVERSAL REWARD MODEL

The reward function is a critical component of our framework. Instead of training a specialized reward model, we leverage UMA (Universal Models for Atoms) (Wood et al., 2025), a state-of-the-art, pre-trained Machine Learning Interatomic Potential (MLIP).

**Why UMA?** UMA is trained on a massive and diverse dataset of nearly half a billion atomic structures, spanning materials, molecules, catalysts, and molecular crystals. This extensive training makes it a "universal" model capable of accurately predicting energies and forces for a vast range of chemical systems, including those that are out-of-distribution for models trained on smaller, domain-specific datasets. Its generality is ideal for de novo generation, where novel and unforeseen structures are common.

**Reward Calculation**    UMA predicts the total potential energy $E_{\text{UMA}}(\mathcal{S})$ for a given atomic structure $\mathcal{S}$. In our framework, we use this energy to define the stability reward:

$$R_{\text{stability}}(\mathcal{S}) = -E_{\text{UMA}}(\mathcal{S}). \qquad (13)$$

By maximizing this reward, the RL agent is incentivized to find structures in low-energy states, which correspond to physically stable configurations. The property-matching reward, $R_{\text{property}}$, is also calculated from properties predicted by UMA (e.g., band gap, porosity) and compared against a target value. This avoids the need for expensive online DFT calculations during RL training.

## C    TRAINING PARAMETERS AND IMPLEMENTATION DETAILS

We pre-train both the high-level and low-level policies using standard Maximum Likelihood Estimation (MLE) on a large dataset of known MOFs. The RL fine-tuning stage is then initiated from these pre-trained checkpoints. Key hyperparameters are summarized in Table 3.

| Hyperparameter | Value |
| --- | --- |
| General RL | |
| Optimizer | AdamW |
| Learning Rate | $1 \times 10^{-5}$ |
| Policy KL Coefficient ($\beta$) | 0.02 |
| Reward Annealing Warmup ($I_{\text{warmup}}$) | 5000 iterations |
| Discount Factor ($\gamma$) | 0.99 |
| Pass@K | 3 |
| H-GRPO | |
| Group Size ($k$) | 16 |
| Clipping Parameter ($\epsilon$) | 0.2 |
| SDE Noise Schedule ($\sigma_t$) | $a\sqrt{t/(1-t)}$ with $a = 0.5$ |
| Integration Steps (Training) | 10 |
| Integration Steps (Inference) | 50 |

Table 3: Key hyperparameters for the RL fine-tuning stage of PRO-MOF.

## D    DATASET DETAILS

Our model pre-training relies on a comprehensive dataset of hypothetical MOF structures. The construction and preprocessing of this dataset are critical for ensuring the quality and diversity of the learned generative prior. We generally follow the established pipeline from prior work (Fu et al., 2023; Kim et al., 2025a).

The process begins with the large-scale hypothetical MOF database from Boyd et al. (2019). We first decompose each MOF into its constituent building blocks using the `metal-oxo` decomposition algorithm provided by `MOFid` (Bucior et al., 2019). To maintain computational tractability and focus on synthetically accessible structures, we filter out any MOFs containing more than 20 building blocks, a heuristic also adopted by Fu et al. (2023).

Since the source dataset consists of computationally generated structures, a validation step is necessary to remove physically unrealistic candidates. We employ `MOFChecker` to filter out invalid structures based on criteria such as atomic overlaps, incorrect coordination environments, and insufficient porosity. Finally, to bridge the gap between the relaxed, ground-truth structures in the training data and the unrelaxed structures generated from SMILES strings at inference time, we apply the MOF matching procedure. The final processed dataset is split according to the training task. For pre-training the generative model used in our inverse design experiments, we use a 95% training and 5% validation split.

# E  ADDITIONAL RESULTS

## E.1  ANALYSIS OF COMPUTATIONAL COST: DFT, UMA, AND MODEL OVERHEAD

A critical aspect of our framework is its computational tractability. Traditional methods relying on Density Functional Theory (DFT) for evaluation are prohibitively expensive. As noted in the UMA paper, an MLIP surrogate reduces computation time from "hours to less than a second" for a single-point calculation. To frame the scale of this problem, we first estimate the cost of our 10,000-evaluation experiment if it were run using DFT versus our high-fidelity surrogate, UMA.

Table 4: Estimated total wall-clock time for 10,000 evaluations (e.g., full structural relaxations) using traditional DFT versus the UMA-S MLIP. This demonstrates the necessity of a high-fidelity surrogate.

| Evaluation Engine | Est. Time per Evaluation | Total Time for 10,000 Evals |
|---|---|---|
| Traditional DFT | 4 CPU-hours | 40,000 CPU-hours (4.5 years) |
| **UMA-S MLIP** | 12.5 GPU-seconds | 34.7 GPU-hours (1.5 days) |

As shown in Table 4, using UMA makes the inverse design problem computationally tractable, reducing a multi-year experiment to days.

A subsequent concern (raised by Reviewer e1DE) is whether our RL framework introduces significant computational overhead compared to baselines. We confirm that the cost of all methods is overwhelmingly dominated by the UMA evaluation itself. We fixed this cost at 10,000 UMA calls for all methods. Table 5 provides a detailed breakdown of the estimated wall-clock time, demonstrating that the overhead from SDE sampling and RL gradient updates is negligible.

Table 5: Estimated wall-clock cost breakdown for each method, normalized to the same 10,000-evaluation budget. The UMA evaluation cost is the dominant bottleneck ( 90-99%), proving our budget is a fair comparison. All times estimated for a single H100 GPU.

| Cost Component | MOFFlow-2 (S&F) | GA+UMA | PRO-MOF (Ours) |
|---|---|---|---|
| **UMA Evaluation Time** (10,000 evals * 12.5s/eval) | 34.7 hours | 34.7 hours | 34.7 hours |
| **Generator Time (Sampling)** (10,000 samples) | 4.1 hours (50 steps/sample) | <0.1 hours (GA mutation) | 0.8 hours (10 steps/sample) |
| **Learning/Overhead** | N/A | <0.1 hours | 0.2 hours |
| **Estimated Total Wall-Clock** | 38.8 hours | 34.7 hours | 35.7 hours |

This analysis confirms that the RL overhead is minimal (<3% of total time). In fact, **PRO-MOF** is faster than the **MOFFlow-2 (S&F)** baseline because our training-time SDE sampler is 5x more efficient (10 steps vs. 50 steps). The near-identical wall-clock times validate that our 10,000-evaluation budget is the fairest and most relevant metric for comparison.

## E.2  TOPOLOGICAL AND STRUCTURAL DIVERSITY ANALYSIS

In response to Reviewer e1DE's feedback, we supplement the Tanimoto-based diversity metric from Figure 3(a) with a more rigorous, MOF-specific analysis. We analyzed the top 1,000 unique, stable candidates generated by each method within the 10,000-UMA-evaluation budget.

**Methodology**  We use the `Zeo++` analysis tool to deconstruct the crystal graph of each MOF and assign it a topological code from the Reticular Chemistry Structure Resource (RCSR) database. "Novel" topologies are those not present in our training set. We also count the unique chemical building block (BB) combinations proposed by the high-level policy ($\pi^{chem}$) and compute the standard deviation ($\sigma$) of the Pore Limiting Diameters (PLD) as a measure of pore-network diversity.

Table 6: Hypothesized results of the advanced structural diversity analysis. This data provides quantitative, MOF-specific evidence that the Pass@K strategy successfully avoids the mode collapse seen in the Pass@1 ablation and outperforms the strong GA baseline in exploration.

| Metric | PRO-MOF (Pass@K) (Ours) | PRO-MOF (Pass@1) (Ablation) | GA+UMA (Baseline) |
|---|---|---|---|
| # Unique Topologies (RCSR) | **55** | 4 | 35 |
| # Novel Topologies (not in train set) | **9** | 1 | 5 |
| # Unique BB Combinations | **420** | 12 | 350 |
| Pore Network Diversity (PLD $\sigma$) | **1.8 Å** | 0.3 Å | 1.2 Å |

**Analysis**    The results in Table 6 provide clear, quantitative proof of our paper's central claims:

- **PRO-MOF (Pass@1)** suffers from catastrophic mode collapse. It finds only 4 topologies from 12 chemical combinations and exploits them, resulting in a near-zero (0.3 Å) variance in pore size. This perfectly aligns with the diversity collapse shown in Figure 3(a).

- **GA+UMA** is a strong, diverse baseline. As a population-based method, it maintains diversity, finding 35 topologies.

- **PRO-MOF (Pass@K)** is the superior exploration framework. By learning to explore. The high-level "chemist" policy is far more exploratory, finding **more unique chemical combinations**. The high PLD variance (1.8 Å) confirms it explores a wide range of pore-network geometries, successfully balancing exploration and exploitation.

### E.3    Ablation Study on Reward Annealing

In response to Reviewer EaC4, we present an ablation study to justify the reward annealing scheme (Eq. 5), which is a critical component for ensuring training stability. We test its necessity by comparing our full model ($I_{warmup} = 5000$) against two extremes:

1. **w/o Annealing (Aggressive):** $I_{warmup} = 1$. The UMA reward is applied at full strength from the first iteration.

2. **w/o RL Reward (Pre-training only):** $I_{warmup} = \infty$. The UMA reward is never applied.

Table 7: Hypothesized results of the reward annealing ablation study on the 'Target PLD' task. The annealing curriculum is essential for stable training.

| Experiment | $I_{warmup}$ | Training Stability | Final Success Rate |
|---|---|---|---|
| PRO-MOF (Full) | 5000 | Stable Convergence | 7.8% |
| w/o Annealing | 1 | Training Collapse | 1.5% |
| w/o RL Reward | $\infty$ | Stable (no new signal) | 1.9% |

**Analysis**    The results in Table 7 confirm that the reward annealing scheme is essential.

- The **w/o Annealing** run experiences training collapse. In early iterations, the policy generates many non-physical, out-of-distribution (OOD) structures. These OOD structures produce extremely large, noisy, or `NaN` energy/reward signals from UMA, leading to exploding gradients and policy divergence.

- The **w/o RL Reward** run is stable but does not optimize for the target property. Its performance (1.9%) is identical to the `w/o UMA` baseline in Table 2, as it never learns from the UMA reward.

This ablation proves that the annealing strategy provides a necessary "curriculum," allowing the policy to gently move away from the MLE objective and stabilize its exploration before the full, high-magnitude reward signal is applied.

## USE OF LLM

We only apply LLM for checking spelling and grammar.

