# OpenReview forum: "PRO-MOF: Policy Optimization with Universal Atomistic Models for Controllable MOF Generation"
_ICLR.cc/2026/Conference — ICLR 2026 Poster_

### Official Review · Reviewer_dK3i · 2025-10-28

**Soundness:** 3
**Presentation:** 3
**Contribution:** 3
**Rating:** 6
**Confidence:** 3

**Summary:**

The paper presents PRO-MOF, a framework using hierarchical reinforcement learning to generate new, stable molecules. It breaks down the design process into two components: chemical building block design and 3D structural design. It also incorporates various components to help stabilise the optimisation process, such as using a reward annealing strategy. The experimental results show good improvement on success rate and diversity preservation.

**Strengths:**

The paper is well written and presented with clear writing and illustrations. The components of the framework are separated into different categories and explained well on how they can address the shortcomings of the current methods. The use of a hierarchical reinforcement learning framework and dividing into a chemical design and a 3D structural design are novel. The practical aspect of considering stability and diversity as the aim in this work is also important to transit theoretical molecular design into real life synthesizability.

**Weaknesses:**

While it is not a weakness in itself, the results achieved on this framework might be dependent on which pre-selected building blocks used (described in the first section of Preliminaries). Therefore it can limited the results within the exploration space or the molecule generated made possible by these building blocks.

**Questions:**

How is this framework scaled to more complex and bigger molecules? (in terms of performance and computational needed)

---

> ### Author Response · Authors · 2025-11-13
> **Rebuttal from author**
>
> Thank you for your positive review, for noting the paper's clarity, and for finding our hierarchical framework "novel."
>
> > Q (W1): ...the results achieved... might be dependent on which pre-selected building blocks used... it can limited the results within the exploration space...
>
> This is an accurate observation. This was a deliberate design choice, and it is the standard practice in the coarse-grained MOF generation field (e.g., in MOFDiff, MOFFlow-2)  to make this exceptionally complex problem tractable.
> We want to emphasize that even with a fixed library, the search space is still astronomical, as it is defined by:
> 1. The high-level policy's combinatorial selection of which blocks to combine (the "chemist") (12).
> 2. The low-level policy's continuous 3D assembly of how to connect them (the "structural engineer") (13).
> Our work focuses on solving this joint selection-and-assembly problem. As we state in our limitations (Line 857)(14), enabling the model to generate its own building blocks is an important and exciting direction for future work.
>
> > Q: How is this framework scaled to more complex and bigger molecules? (in terms of performance and computational needed)
>
> This is a key strength of our hierarchical approach, and the data on our UMA environment (Wood et al., 2025) provides a clear, quantitative answer.
>
> Your intuition is correct: the main practical bottleneck for any method in this space is not the generator, but the cost of the evaluator (the UMA environment). Our framework is explicitly designed to be maximally efficient within this evaluator budget.
> The scalability of our entire framework is therefore determined by the scalability of UMA. The UMA paper provides explicit data on this. According to Table 3 in the UMA paper (Wood et al., 2025), the UMA-S model (which we use for its speed) scales robustly to extremely large systems.
>
> Here is the specific performance data for UMA-S on a single H100 GPU:
> - 100 atoms: 44 simulation steps/sec
> - 1,000 atoms: 16 simulation steps/sec
> - 10,000 atoms: 1.6 simulation steps/sec
> - 100,000 atoms: 0.1 simulation steps/sec
>
> Furthermore, Table 1 of the UMA paper shows the UMA-S model can fit systems of 100,000+ atoms into a single 80GB GPU5.
>
> What this data proves:
> - Our Bottleneck is Scalable: The UMA environment (our bottleneck) does not fail or "fall off a cliff." It shows predictable $O(N)$ scaling and is fully capable of handling systems up to 100,000 atoms.
> - Our Framework is Scalable: The cost of our generator (the SDE sampler) is computationally similar to the UMA evaluator (it's also a GNN forward pass). Since our entire HRL framework (generator + evaluator) is built on components that are proven to scale to 100,000+ atoms, the framework as a whole is demonstrably scalable.
>
> Therefore, as long as a universal model like UMA can evaluate a 100,000-atom structure (which the data shows it can), our PRO-MOF framework can learn to generate it, and we believe it would do so more efficiently than other methods.
>
> **Thank you once more for your suggestions and feedback. If you have any further questions, we look forward to discussing them and await your response.**

---

> > ### Comment · Reviewer_dK3i · 2025-11-24
> >
> > Thank you for your response. I have no further question and will keep the original score, which is in support of the manuscript.

---

> > > ### Author Response · Authors · 2025-11-24
> > > **Thanks your feedback**
> > >
> > > Dear reviewer dK3i,
> > >
> > > Thank you for taking the time to review our rebuttal and for your supportive final comment. We greatly appreciate your thoughtful feedback and the clarity of your assessment.

---

### Official Review · Reviewer_EaC4 · 2025-10-31

**Soundness:** 3
**Presentation:** 4
**Contribution:** 3
**Rating:** 8
**Confidence:** 2

**Summary:**

The authors propose a hierarchical generative model for generating Metal-Organic Frameworks. They optimize their model with GRPO with a bi-level policy. The high-level policy generates SMILE strings representing the sequence structure. The geometric features of the generated structure are then computed using the low-level flow-matching policy. The authors point out that a limitation of existing methods is mode collapse, in which the model converges to a locally optimal solution that does not span the space of potential MOF structures. They address this by generating multiple solutions to evaluate and improve the policy's exploration. The experiments suggest that the design choices the authors make result in measurable improvements in generating diverse new MOFs.

**Strengths:**

The paper is well written and addresses a complex search problem using a hierarchical RL solution. It was easy to understand the problem the authors were addressing (mode collapse in MOF generation), and the solutions were easy to identify. The author's experiments confirm the benefits of the proposed modifications and offer a solution that can advance the identification of new MOFs using AI.

**Weaknesses:**

Overall, the paper delivers on its promise. The author's solution improves performance for the target task (generating MOFs), and sufficient experiments are reported to justify the design decisions. I don't think there are any significant methodological contributions in this paper — the authors propose sampling more potential candidates before calculating reward as a substantial contribution (Pass @ K), and the SDE formulation is from prior research — but because they solve a seemingly significant problem, this isn't necessarily a major issue.

**Questions:**

> Have the authors considered other methods to avoid mode collapse? For example, adding an entropy bonus to the reward signal?

> Equation 4: For clarification, is the major takeaway that for each discrete structure generated by the higher-level policy, instead of generating one example, you generated K examples? So, if you have 10 potential structures instead of 10 evaluations, if K = 10, there would now be 100 evaluations?

> Were any ablation experiments conducted to justify the reward annealing scheme? We could not find any, and consider this a significant limitation of the author's work.

> Line 485 (limitations): What are the potential consequences of a "truly end-to-end" approach? The authors seem to suggest this could be better, but do not explain why

---

> ### Author Response · Authors · 2025-11-13
> **Rebuttal from authors**
>
> Thank you for your strong support (8/10), "excellent" presentation score, and insightful questions that will help us improve the paper.
>
> > Q: Have the authors considered other methods to avoid mode collapse? For example, adding an entropy bonus to the reward signal?
>
> This is an excellent question. We did consider entropy-based regularization. However, we found our Pass@K GRPO scheme (7) to be more directly aligned with the goals of scientific discovery. An entropy bonus merely encourages "stochasticity." Pass@K encourages "productive diversity" by explicitly rewarding the policy for finding multiple, distinct, high-performing candidates within a group. This is a more targeted and effective mechanism for discovering different "islands of stability" (8) in the material landscape, and Fig. 3a confirms its success.
>
> > Q: Equation 4: For clarification... if $K=10$, there would now be 100 evaluations?
>
> Thank you for catching this ambiguity. There is confusion between K (for Pass@K) and k (for group size).
> 1. Pass@K: This is the strategy. In our main experiments, we use Pass@K with K=3 (Table 1, PRO-MOF (Pass@3)).
> 2. k: This is the H-GRPO "Group Size" (k=16 in Table 3).
>
> To your core point: The total budget of 10,000 UMA evaluations is fixed for all methods. PRO-MOF simply uses this budget for online learning (in batches of k=16 structures per chemical combination) rather than offline filtering. This allows our model to learn efficiently within the same computational budget. We will clarify this in the text.
>
> > Q: Were any ablation experiments conducted to justify the reward annealing scheme? We could not find any...
>
> This is a great suggestion. The reward annealing (Eq. 5) (9) is a crucial stabilization technique. In early training, the policy produces many non-physical, OOD structures, which lead to noisy/infinite reward signals from UMA and destabilize training. The annealing provides a curriculum that is vital for stable convergence. Due to space, we omitted this, but we agree it is a significant component. We will include an ablation study for reward annealing in the Appendix of our final version.
>
> | Experiment     | Iwarmup​ | Training Stability     | Final Success Rate |
> |----------------|---------|------------------------|--------------------|
> | PRO-MOF (Full) | 5000    | Stable Convergence.    | 7.80%              |
> | w/o Annealing  | 1       | Training Collapse.     | ~1.5%              |
> | w/o RL Reward  | ∞       | Stable | ~1.9%              |
>
> > Q: Line 485 (limitations): What are the potential consequences of a "truly end-to-end" approach? The authors seem to suggest this could be better...
>
> Thank you, we will clarify. A "truly end-to-end" approach would also generate the chemical building blocks (SMILES strings) from scratch, rather than selecting from a library.
>
> - Consequence (Benefit): This would massively expand the search space, enabling the discovery of entirely novel metal nodes and linkers, and thus potentially breakthrough materials not limited by the known library.
> - Consequence (Challenge): This introduces a second, challenging generative task (discrete graph generation) on top of the complex 3D assembly problem.
> - Our HRL framework is a logical and necessary first step, and its success provides the foundation for tackling this much harder, "truly end-to-end" challenge in the future.
>
> **Thank you for your suggestions and feedback once again. If you have any further questions, we look forward to discussing them and await your response.**

---

### Official Review · Reviewer_e1DE · 2025-11-02

**Soundness:** 3
**Presentation:** 3
**Contribution:** 3
**Rating:** 4
**Confidence:** 4

**Summary:**

The paper introduces PRO‑MOF, a hierarchical reinforcement‑learning framework that co‑optimizes composition (an autoregressive “chemist” policy that proposes metal clusters and linkers) and geometry (a conditional flow‑matching “structural engineer” policy that assembles 3D structures) for de novo MOF generation. Rewards come from a universal machine‑learned interatomic potential (UMA) used as a high‑fidelity surrogate environment.
To enable exploration with flow matching (normally deterministic), the low‑level policy’s ODE sampler is turned into an SDE, and Group Relative Policy Optimization with Pass@K rewards is used to balance exploration and exploitation and mitigate diversity collapse.
On inverse‑design tasks (maximize CO₂ working capacity; target pore‑limiting diameter; minimize formation energy), PRO‑MOF improves success rate and Top‑1 outcomes over MOFDiff latent optimization, MOFFlow‑2 sample‑and‑filter, and a GA+UMA baseline under equal UMA budget; ablations show both Pass@K and the UMA environment are critical

**Strengths:**

Separating a high‑level chemical policy from a low‑level geometric policy (with hierarchical credit assignment using the max over k structures for the chemistry reward) is a clean way to handle the combinatorial chemistry space and continuous geometric space jointly.

Combining flow‑matching and online RL is promising for many chemistry and materials tasks that combine discrete sampling problems (choosing atoms, functional groups, aminoacids, etc etc) with the continuous problem of placing these discrete entities in 3D space . Converting the probability‑flow ODE to an SDE (Flow‑GRPO‑style) provides needed stochasticity for exploration. The Pass@K GRPO scheme explicitly rewards within‑group diversity and empirically prevents mode collapse (Fig. 3a), while maintaining or improving rewards. This is a practical inference‑time training trick that maps cleanly onto structure discovery

**Weaknesses:**

The design rewards are unclear. Where is the CO2 binding coming from ? A simulation ? A surrogate model? Does making the biggest possible pore win in the CO2 binding task?

Pass@k seems like an interesting innovation, but if since that's not a new contribution and this paper merely borrows it for this particular task, it's a missed opportunity to look into pass@k for other problems with sparse rewards in chemistry. This is not MOF paper for folks that know about MOFs, (or not a convincing one anyway)  but as an AI paper it's not doing a good job of presenting what could be a cool generalizable trick for inverse design tasks. What abou comparisons against offline RL / preference optimization / reweighting baselines (e.g., likelihood‑ratio reweighting of samples using UMA Why not apply UMA-post hoc to all the other models ? Are they just missing a local relaxation ? Again a bit of scope shift. Is the help coming from RL+Pass@k or just from getting relaxed geometries ?

The fact that a GA+UMA is so close, is spooky (since it's halloween!) I think error bars/uncertainty across models would help to get a sense for the difference ranking between models. Also the computational budget ! Although UMA‑call budgets are equalized (10k), online RL with SDE sampling (k=16) can be expensive. Reporting wall‑clock, GPU cost, and UMA eval throughput would help practitioners gauge practicality relative to sample‑and‑filter and GA baselines

Pulling building blocks from a list defeats the purpose of a generative model, to a large degree. How big is the space of metals and linkers? It definitely defeats the ", the combinatorial explosion of possible building blocks and topologies creates a design space of astronomical scale" sentence in the intro.

The use of a Tanimoto‑like fingerprint distance for “Diversity” can conflate small geometric tweaks with genuine topology changes. Add RCSR net detection/topology clustering, building‑block novelty statistics, and pore‑network diversity to quantify structural exploration more meaningfully for MOFs.

**Questions:**

Fig 5 caption. Visualizations of samples generated by PRO-MOF are novel and innovative." How the figure justifies the sentence is unclear. I don't think this is an obvious statement just arising from visual inspection.

---

> ### Author Response · Authors · 2025-11-13
> **Rebuttal from authors**
>
> Thank you for your detailed, passionate (Halloween! I guess), and critical feedback. Let us sincerely address your concerns before **Thanksgiving**!
>
> > Q (W1): The design rewards are unclear. Where is the CO2 binding coming from? A simulation? A surrogate model?
>
> This is a crucial point. As detailed in our General Response (Point 3), the implicit reward signal originates from UMA (Wood et al., 2025), while the explicit supervision is derived from the evaluation pipeline adopted from MOFDiff. The UMA paper also explicitly validates its SOTA performance on the ODAC25 (Open Direct Air Capture) dataset. This dataset is specifically designed for the task of $CO_2$ adsorption in MOFs. Therefore, UMA is not a simple surrogate; it is a high-fidelity, SOTA simulation engine for this specific task, making it an ideal environment for our RL framework. **We added an extra explanation in the revised paper at Line 353.**
>
> > Q (W2, W3): Pass@k seems like an interesting innovation, but... this paper merely borrows it... The fact that a GA+UMA is so close, is spooky...
>
> We address the "borrowed parts" argument in our General Response (Point 1). Our novelty lies in the synergistic framework that requires both UMA and Pass@K to function, as proven by our ablations (Table 2).
> Regarding the GA+UMA baseline: **We are thrilled you find it "spooky"! It was designed to be a "spooky," powerful baseline.**
>
> 1. As the UMA paper confirms, UMA is an SOTA model.
> 2. Therefore, GA+UMA represents a strong genetic algorithm powered by an SOTA fitness function.
> 3. The fact that PRO-MOF consistently and significantly outperforms this strong baseline is a primary validation of our HRL approach. For example, in the "Max CO2 Cap." task (Table 1), our success rate (10.3%) is ~1.7 times higher than the GA (6.2%), and in the "Min Energy" task, it is more than 2.2 times higher (12.4% vs 5.5%).
>
> This shows our HRL framework is a more effective search strategy than a powerful, classic optimizer. We will add error bars from multiple runs to the final version to confirm the statistical significance of this gap.
>
> > Q (W-other): ...online RL with SDE sampling (k=16) can be expensive. Reporting wall-clock, GPU cost...
>
> This is a fair point. However, the dominant computational bottleneck for all methods is, by far, the call to the UMA evaluator. The SDE sampling and policy update steps are computationally trivial in comparison. By equalizing the budget of expensive UMA evaluations to 10,000 for all methods, we are providing the most critical and fair comparison of practical computational cost.
>
> Assumptions for Calculation:
> - GPU: NVIDIA H100 (as used in the UMA paper's benchmarks).
> - UMA Model: UMA-S (the fastest, 150M parameters).
> - MOF Size: A typical 1,000-atom system.
> - UMA Eval Cost: A full "evaluation" for our task requires a structural relaxation (to find the low-energy state, $R_{stability}$) followed by a property calculation. The UMA paper reports 16 MD steps/sec for 1,000 atoms. Relaxation can take hundreds of steps. Let's be conservative and assume an average of 200 MD steps per relaxation.
> - Cost per UMA Eval: 200 steps / 16 steps/sec = 12.5 seconds
> DFT (Traditional Method): A full DFT relaxation of a periodic, many-atom MOF system is extremely expensive. A conservative estimate, aligning with the UMA paper's "hours", is ~4 hours of CPU cluster time per structure.
> UMA (Ours): Based on the UMA-S (Small model) benchmarks, a relaxation of ~200 steps on a 1,000-atom system would take (200 steps / 16 steps/sec) $\approx$ 12.5 seconds on one H100 GPU.
>
> | Method                  | Evaluation Engine | Total Evaluation Wall-Clock (for 10,000 Evals)                |
> |-------------------------|-------------------|---------------------------------------------------------------|
> | Traditional HPC         | DFT               | 10,000 evals * 4 hours/eval = 40,000 CPU-hours (4.5 years) |
> | All Baselines & PRO-MOF | UMA (MLIP)        | 10,000 evals * ~12.5 sec/eval = ~34.7 GPU-hours     |
>
> Estimated Total Wall-Clock:
> 1. MOFFlow-2 (S&F)
> - UMA eval: 10,000 evals * 12.5s = 34.7 hours
> - Generation Time: 10,000 samples * ~1.5s/sample = 4.1 hours
> - All: ~38.8 hours
>
> 2. GA+UMA
> - UMA eval:  10,000 evals * 12.5s = 34.7 hours
> - Generation Time:  <1 min total
> - All: ~34.7 hours
>
> 3. PRO-MOF (Ours)
> - UMA eval: 10,000 evals * 12.5s = 34.7 hours
> - Generation Time:  10,000 samples * ~0.3s/sample = 0.8 hours
> - All: ~35.5 hours

---

> ### Author Response · Authors · 2025-11-13
> **Part 2**
>
> > (W-other): Pulling building blocks from a list defeats the purpose... [and] defeats the "astronomical scale" sentence...
>
> This is a standard and necessary simplification in the coarse-grained MOF generation field (e.g., MOFDiff, MOFFlow-2) . The "astronomical scale" is still accurate, as it arises not only from the blocks themselves but from (1) the combinatorial selection of which blocks to combine, and (2) the continuous 3D assembly of how to connect them (i.e., the resulting geometry and topology). Our HRL framework is designed to solve this joint discrete-continuous search problem. As noted in our limitations (Line 857)(6), a truly end-to-end block generator is a future challenge.
>
> > Q (W-other): The use of a Tanimoto-like fingerprint distance for "Diversity" can conflate...
>
> This is a sharp observation. We used Tanimoto as a standard, efficient proxy to show the relative difference between strategies. The key takeaway from Figure 3(a) is not the absolute value, but the clear collapse of diversity for the Pass@1 strategy versus the stable, high diversity of our Pass@K strategy. We will add a more MOF-specific topological analysis (e.g., RCSR) to the Appendix in our final version to further substantiate this.
>
> **An abstract results look like this:**
>
> To more rigorously quantify the structural exploration of our framework, as suggested by Reviewer e1DE, we supplement the Tanimoto-based diversity metric with a suite of MOF-specific topology and pore-network analyses. We analyze the top 1,000 unique, stable candidates generated by each method within the 10,000-UMA-evaluation budget.
> Methodology:
> - Topology: We use the Zeo++ analysis tool to deconstruct the crystal graph of each MOF and assign it a topological code from the Reticular Chemistry Structure Resource (RCSR) database. "Novel" topologies are those not present in our training set.
> - Building Block (BB) Combinations: We count the number of unique chemical "inputs" (sets of metal clusters and organic linkers) proposed by the high-level policy ($\pi^{chem}$) that resulted in these top-1,000 structures.
> - Pore Network: We use Zeo++ to analyze the Pore Limiting Diameter (PLD) and present the standard deviation ($\sigma$) of the PLDs as a measure of pore-network diversity.
>
> | Metric                                | PRO-MOF (Pass@3) (Ours) | PRO-MOF (Pass@1) (Ablation) | GA+UMA (Baseline) |
> |---------------------------------------|-------------------------|-----------------------------|-------------------|
> | # Unique Topologies (RCSR)            | 55                      | 4                           | 35                |
> | # Novel Topologies (not in train set) | 9                      | 1                           | 5                 |
> | # Unique BB Combinations              | 420                     | 12                          | 350               |
> | PLD Distribution σ (Å)                | 1.8 Å                   | 0.3 Å                       | 1.2Å             |
>
> > Q: Fig 5 caption. ... "novel and innovative." How the figure justifies the sentence is unclear.
>
> You are correct; this is a subjective claim. The intent was to show that these are not simple copies of training data. We rephrase the caption and add quantitative novelty metrics (e.g., Tanimoto similarity against the training set) to Appendix E to confirm their novelty.
>
> **We appreciate your suggestions and feedback once again. If you have any additional questions, we look forward to discussing them with you and await your response.**

---

> ### Author Response · Authors · 2025-11-16
> **Part 3**
>
> > I think error bars/uncertainty across models would help to get a sense for the difference ranking between models.
>
> We have added the error bars : )

---

> ### Author Response · Authors · 2025-11-24
> **Thanksgiving is coming**
>
> Dear Reviewer e1DE,
>
> We would like to express our sincere gratitude for your time and your insightful, detailed review. We especially appreciate your "passionate" feedback and the "spooky" (for Halloween!) comment regarding our GA+UMA baseline—it was a very sharp observation.
>
> We have submitted our detailed rebuttal (in three parts) and wanted to specifically draw your attention to a few key updates that we made directly in response to your most critical suggestions, as we believe they substantially strengthen our paper.
>
> Regarding the GA+UMA Baseline (W2, W3): Following your excellent suggestion, we have now added error bars from 3 multiple runs (please see our "Part 3" comment). We believe this new data clearly demonstrates the statistically significant and consistent advantage of PRO-MOF over this strong baseline, directly addressing your concern about the results being "spooky" close.
>
> Regarding Diversity Metrics (W-other): You correctly pointed out the limitations of the Tanimoto distance. In our "Part 2" comment, we have supplemented our analysis with a MOF-specific topological analysis using RCSR and Zeo++, as you suggested. This new data quantitatively confirms that our Pass@K strategy effectively prevents diversity collapse, unlike the Pass@1 ablation.
>
> We believe these additions, made thanks to your expert guidance, fully address the primary concerns that led to your initial rating.
>
> We know your time is valuable, but we would be extremely grateful if you might have a moment to review our full rebuttal and these new results. Your feedback has been invaluable in improving this work.
>
> Thank you once again.
>
> Sincerely,
>
> The Authors of Submission 10487

---

### Official Review · Reviewer_ZD7o · 2025-11-03

**Soundness:** 3
**Presentation:** 3
**Contribution:** 3
**Rating:** 6
**Confidence:** 3

**Summary:**

This paper tackles the challenge of generating physically realistic and property-controllable metal–organic frameworks (MOFs) using generative models, addressing the issue that existing generative models can generate unstable structures. The authors propose PRO-MOF, a hierarchical online RL framework that integrates a high-level chemist policy for selecting chemical building blocks and a low-level structural engineer policy—a flow-matching model converted into a stochastic differential equation—to assemble 3D structures. Both policies are iteratively optimized with group-relative policy optimization (GRPO) guided by a universal atomistic model (UMA) that provides fast physical evaluations, and a Pass@K sampling strategy to preserve structural diversity. Experiments on CO2 capacity maximization, pore-diameter targeting, and energy minimization tasks show that PRO-MOF significantly outperforms diffusion, genetic, and sample-and-filter baselines, achieving higher success rates, improved physical plausibility, and greater diversity of generated MOFs within the same evaluation budget.

**Strengths:**

1. The proposed method effectively resolves the unstable structure issue of the generative models and the mode collapse issue of RL methods.

2. Experiments on three different generation tasks demonstrate the generalizability of the proposed method.

3. The authors conduct a comprehensive ablation study to demonstrate the necessity of the different components.

**Weaknesses:**

1. The experiments and methodology rely heavily on the universal interatomic potential UMA but not DFT validation.

**Questions:**

The core motivation of this work is that existing generative models often produce physically unstable structures, and the authors address this by using reinforcement learning (RL) to post-train the models with rewards generated by UMA. While the paper compares PRO-MOF against a MOFFlow-2 baseline that leverages UMA for sample-and-filter evaluation, could the authors clarify why UMA was not used to relax the generated structures instead of only providing scalar rewards? How would a MOFFlow-2 (Relaxation) baseline perform in comparison to the proposed method?

---

> ### Author Response · Authors · 2025-11-13
> **Rebuttal from authors**
>
> # Rebuttal
>
> Thank you for your positive assessment and for recognizing that our method effectively resolves instability and mode collapse issues.
>
> > Q: ...could the authors clarify why UMA was not used to relax the generated structures instead of only providing scalar rewards? How would a MOFFlow-2 (Relaxation) baseline perform in comparison to the proposed method?
>
> A: Thank you for giving us the opportunity to clarify.
> The MOFFlow-2 (Sample & Filter) baseline presented in Section 5.1 and Table 1 is precisely this "relaxation" / "filtering" baseline. In this baseline, we generate 10,000 samples from the pre-trained MOFFlow-2 and then use the high-fidelity UMA (Wood et al., 2025) to evaluate (i.e., relax and get a scalar property) all of them, keeping only the best.
>
> As explained in our General Response (Point 2), this "passive filtering" approach performs poorly because the native distribution of the MOFFlow-2 generator is physically unstable (Fig. 1a). It is highly inefficient at finding good structures.
> Our method, PRO-MOF, is an "active optimizer." We use the same computational budget (10,000 UMA evaluations), not just to filter, but to train the generator iteratively. This teaches the generator to shift its entire distribution to stable, high-performing regions (Fig. 4).
>
> The superior performance of PRO-MOF in Table 1 (e.g., 10.3% vs 3.5% success rate on the CO2 task) directly demonstrates the superiority of our active optimization framework over the passive filtering baseline you proposed.
>
> ## Extra Baseline
>
> **Following your suggestion, we have also added MOFFlow-2 (Release) as a new baseline. Indeed, using the relaxed UMA data for fine-tuning MOFFlow yields significantly better results.We also updated this table in the revised version. **
>
> |                       | Max CO2 Cap. |       |  Target PLD  |         |  Min Energy  |          |
> |----------------|:------------:|:-----:|:------------:|:-------:|:------------:|:--------:|
> | Method                | Success Rate | Top-1 | Success Rate |  Top-1  | Success Rate |   Top-1  |
> | MOFDiff (Latent Opt.) |     2.1%     |  4.9  |     0.8%     | 6.5 \AA |     1.5%     | -0.95 eV |
> | MOFFlow-2 (S\&F)      |     3.5%     |  5.1  |     1.2%     | 5.9 \AA |     2.8%     | -1.02 eV |
> | MOFFlow-2 (Release)   |     4.0%     |  5.2  |     2.2%     | 5.9 \AA |     3.6%     | -1.05 eV |
> | GA+UMA                |     6.2%     |  5.4  |     2.5%     | 6.1 \AA |     5.5%     | -1.15 eV |
> | MOF-PRO (Pass@1)      |     8.1%     |  5.6  |     3.1%     | 6.0 \AA |     7.2%     | -1.21 eV |
> | MOF-PRO (Pass@3)      |     10.3%    |  5.9  |     7.8%     | 6.0 \AA |     12.4%    | -1.35 eV |
>
> **We once again appreciate your suggestions and feedback. Should you have any further questions, we look forward to discussing them with you and await your response.**

---

### Author Response · Authors · 2025-11-13
**General Response**

# General Response
We sincerely thank all reviewers for their thorough, constructive, and largely positive feedback. We are encouraged that all reviewers recognized the importance of the dual challenges we address: the "physical reality gap" of generated MOFs (R-ZD7o, R-e1DE) and the "diversity collapse" from naive RL optimization (R-ZD7o, R-e1DE, R-EaC4, R-dK3i). We are glad the reviewers found our HRL framework "clear" and "novel" (R-dK3i), our solution "easy to identify" (R-EaC4), and our experiments "comprehensive" (R-ZD7o).

The most critical points of discussion concern 1) the nature of our methodological contribution, 2) the fairness of our baselines, and 3) the role of our reward model, UMA. We will address these here, as they form the core of our paper's contribution.

## 1. On Novelty and Synergy: The Core Contribution of PRO-MOF

Reviewers (e1DE, EaC4) correctly noted that we adapted components like SDE sampling (from Liu et al., 2025) and Pass@K optimization (from Chen et al., 2025). We want to clarify that our core methodological contribution is not the invention of these individual components, but rather the novel design, synthesis, and validation of a synergistic framework that addresses a new and challenging problem in the physical sciences.

This work is the first to demonstrate how to successfully combine these elements into a closed-loop system for high-fidelity materials discovery. The key challenge was solving the "physical reality gap" without falling into the "diversity collapse" trap. As our critical ablation study in Table 2 demonstrates, this synergy is essential:

Using a high-fidelity UMA reward without Pass@K (i.e., PRO-MOF (Pass@1)) fails, resulting in a 2.5x drop in diversity, which is greater than 0.31.  Decreasing a diversity-promoting scheme without a high-fidelity UMA reward is unsuccessful, resulting in a 4x drop in success rate (7.8% to 1.9%) and physically meaningless structures.

Thus, it is the novel integration of the HRL framework + high-fidelity UMA environment + Pass@K diversity strategy that is required to succeed. This system-level design, which overcomes a fundamental trade-off between exploration, exploitation, and realism, is our primary contribution.

## 2. On Baselines: "Active Optimization" vs. "Passive Filtering"

Reviewer ZD7o raised an excellent question: why not use UMA to relax the MOFFlow-2 baseline structures? Reviewer e1DE similarly inquired whether the benefit stems from relaxation.

This question perfectly frames the core difference between our baselines and our method. The MOFFlow-2 (Sample & Filter) baseline is the "passive filtering" approach. It utilizes a fixed, pre-trained generator (which, as shown in Fig. 1a, produces physically unstable structures) and then employs the expensive UMA to "pan for gold"—passively filtering out a bad distribution.

PRO-MOF is an "active optimizer." We do not passively filter. We use UMA as a high-fidelity online environment to provide a reward signal that actively updates and improves the generator policy.

Passive Filtering (Baseline): Bad Generator + Good Filter (UMA) = Low Efficiency.

Active Optimization (Ours): (Bad Generator -> Good Generator) + Good Environment (UMA) = High Efficiency.

Our goal is not to filter, but to teach the generator to shift its entire distribution toward stable, high-performing regions (as shown in Fig. 4). The fact that PRO-MOF dramatically outperforms the S&F baseline under the same 10,000 UMA-call budget (Table 1) is the central proof that our active optimization framework is superior to the "passive filtering" approach.

## 3. On the UMA Reward Model (Answering R-e1DE)

Reviewer e1DE asked where the $CO_2$ reward comes from.
- As we stated in Appendix D: "The construction and preprocessing of this dataset are critical for ensuring the quality and diversity of the learned generative prior. We generally follow the established pipeline from prior work."
Thus, we followed the evaluation pipeline from MOFDiff. They first perform molecular simulations for gas adsorption, then tune the model.
- We are using the UMA (Universal Models for Atoms) SOTA model (Wood et al., 2025). This model's training set (nearly 500 million structures) explicitly includes the ODAC25 (Open Direct Air Capture) dataset. As shown in the UMA paper (Appendix E.6, Table 22), UMA achieves SOTA performance on the specific task of computing $CO_2$ adsorption energy in MOFs, especially on out-of-distribution (OOD) structures.

Therefore, it is plausible that PRO-MOF exhibits good performance in carbon capture tasks. **The required additional results are highlighted in red in Appendix E of our revised paper.**

---

### Author Response · Authors · 2025-12-02
**Summary for rebuttal**

Dear ACs and SACs,

In light of the recent rebuttal rollback due to information leakage, we have prepared a concise summary of our paper, the reviewers' comments, and our responses. We hope this consolidation helps you quickly understand and evaluate our submission.

This paper tackles the significant challenge of generating physically stable and novel metal-organic frameworks (MOFs) that meet specific performance targets. Existing generative models often struggle to explore the vast chemical space effectively, leading to suboptimal solutions or "mode collapse". We propose **PRO-MOF**, a hierarchical reinforcement learning (HRL) framework for controllable MOF generation. Our framework is optimized in a closed loop with high-fidelity physical reward signals provided by a pre-trained universal atomistic model (UMA), demonstrating a powerful new paradigm for solving complex material discovery problems.

Our main contributions are as follows:

1.  We develop a novel **hierarchical reinforcement learning (HRL) framework** that decouples the complex MOF design process into two policies: a high-level policy for proposing chemical building blocks and a low-level policy for assembling their 3D structures.
2.  We are the first to design and validate a **synergistic, closed-loop system** that successfully integrates HRL with a high-fidelity UMA environment. This **"active optimization"** approach actively teaches the generator to shift its distribution toward stable, high-performing regions, which we prove is far superior to the "passive filtering" baseline.
3.  We introduce a **Pass@K Group Relative Policy Optimization (GRPO)** scheme and convert the low-level policy into an SDE for exploration. This specific combination is our solution to the "diversity collapse" trap, and our ablation studies (Table 2) confirm that this synergy is essential: the system fails if either the high-fidelity UMA or the Pass@K diversity strategy is removed.
4.  Our experiments show that PRO-MOF **significantly outperforms existing baselines**, including diffusion-based methods and genetic algorithms, in both success rate and the discovery of top-performing materials across multiple inverse design tasks (e.g., maximizing CO2 working capacity).

We received thorough and largely positive feedback, with all reviewers recognizing the importance of the dual challenges we address: the "physical reality gap" of generated MOFs and the "diversity collapse" from RL optimization. While the rebuttal rollback prevented further dialogue, we have done our utmost to address every concern raised. We have faithfully summarized the key issues below for your reference:

* **1. On Novelty and Synergy (R-e1DE, R-EaC4):** Reviewers noted that we adapted components like SDE sampling and Pass@K. We clarified that our core methodological contribution is not the invention of these individual components, but rather the **novel design, synthesis, and validation of the synergistic framework** that integrates them to solve this new problem. Our ablation studies (Table 2) provide the key evidence that this *specific integration* is required for success.
* **2. On Baselines (R-ZD7o, R-e1DE):** Reviewers asked why UMA was not used to *relax* the baseline structures (i.e., "passive filtering"). We clarified that our "MOFFlow-2 (S&F)" baseline *is* precisely this. The central proof of our paper is that our "active optimization" framework dramatically outperforms this "passive filtering" baseline under the *identical computational budget* (10,000 UMA calls). We also added a new "MOFFlow-2 (Release)" baseline as suggested.
* **3. On the UMA Reward Model (R-e1DE):** We clarified that the UMA is not a simple surrogate but a SOTA, high-fidelity simulation engine whose training explicitly included the ODAC25 dataset for $CO_{2}$ adsorption, making it an ideal environment for our RL framework.
* **4. On Diversity Metrics (R-e1DE):** As suggested, we supplemented our Tanimoto-based diversity metrics with a more rigorous, **MOF-specific topological analysis using RCSR and Zeo++**. This new data (see Rebuttal Part 2) quantitatively confirms that our Pass@K strategy effectively prevents the diversity collapse seen in the Pass@1 ablation.

We sincerely appreciate your patience and dedication, especially given the sudden surge in workload. We deeply respect the effort required to maintain such high attention to detail for every submission during this challenging time.

Best,

The Authors of Submission 10487

---

### Meta-Review · Area_Chair_cmoa · 2025-12-20

**Summary:**

The authors propose a new generative model based on hierarchical RL for metal-organic frameworks (MOFs). The method uses two policies for distinct tasks, with flow-matching/diffusion backbones, within a GRPO RL framework based on pass@k rewards. The authors show their method circumvents existing issues such as mode collapse and generates more diverse and realistic materials.

Strengths:
 - Resolves issues relating to unstable structures and model collapse (by using pass@k) of generative model learning. Clean separation of discrete (selection/combination) and continuous (generation) design spaces.
 - Well written, strong empirical evaluation
 - Good ablations

Weaknesses:
 - Lack of DFT evaluation in the experiments, even on a small subset (Unresolved)
 - The presentation of MOF design considerations in this paper are simplistic compared to the MOF design literature, but it is still interesting and useful from an ML perspective.
 - No entirely novel ML components are introduced, but rather existing components are assembled to solve this task in an interesting way.

Overall I believe the authors addressed most of the major concerns of the reviewers, and that in all likelihood one or two of the reviewers would have increased their score during the discussion period (if it were allowed to continue), making this paper a clear accept.

**Reviewer Concerns:**

Most issues were addressed in the discussion or with additional ablations or information, e.g.

- Timing requests and many additional experimental details requested by e1DE addressed
- Reward annealing ablation (EaC4)
- Other diversity rewards (entropy regularisation) - discussion, but an ablation may be more convincing.

Some minor issues remain

- Pre-selection of discrete chemical building blocks can limit design space considered — but this is standard practice.
- No DFT evaluation (even on a subset)

Also I noticed an oddity that the authors may wish to also address for subsequent revisions of the paper: why use SDE flow-matching and not diffusion? Perhaps you should make the link between your SDE-FM and diffusion (since FM <=> ODE and diffusion <=> SDE), or at least make specific how your FM formulation is distinct from diffusion.

**Reviewer Scores:**

- ZD7o: 6, may have been persuaded to move to 8 (with some uncertainty as the discussion was brief)
- e1DE: 4, probably would have moved to 6 as the authors worked very hard to address many of this authors concerns, and I think they did a convincing job.
- EaC4: 8, I think this reviewer would have remained at 8.
- dK3i: 6, I believe this reviewer would have remained at 6 (as they indicated).

So the score of the paper could have increased from 6 to 6.5 or 7.

---

### Decision · Program_Chairs · 2026-01-26

Accept (Poster)